# A Constant-Factor Bi-Criteria Approximation Guarantee for $k$-means++

**Dennis Wei**
IBM Research
Yorktown Heights, NY 10598, USA
`dwei@us.ibm.com`

## Abstract

This paper studies the $k$-means++ algorithm for clustering as well as the class of $D^\ell$ sampling algorithms to which $k$-means++ belongs. It is shown that for any constant factor $\beta > 1$, selecting $\beta k$ cluster centers by $D^\ell$ sampling yields a constant-factor approximation to the optimal clustering with $k$ centers, in expectation and without conditions on the dataset. This result extends the previously known $O(\log k)$ guarantee for the case $\beta = 1$ to the constant-factor bi-criteria regime. It also improves upon an existing constant-factor bi-criteria result that holds only with constant probability.

## 1   Introduction

The $k$-means problem and its variants constitute one of the most popular paradigms for clustering [15]. Given a set of $n$ data points, the task is to group them into $k$ clusters, each defined by a cluster center, such that the sum of distances from points to cluster centers (raised to a power $\ell$) is minimized. Optimal clustering in this sense is known to be NP-hard [11, 3, 20, 6]. In practice, the most widely used algorithm remains Lloyd's [19] (often referred to as the $k$-means algorithm), which alternates between updating centers given cluster assignments and re-assigning points to clusters.

In this paper, we study an enhancement to Lloyd's algorithm known as $k$-means++ [4] and the more general class of $D^\ell$ sampling algorithms to which $k$-means++ belongs. These algorithms select cluster centers randomly from the given data points with probabilities proportional to their current costs. The clustering can then be refined using Lloyd's algorithm. $D^\ell$ sampling is attractive for two reasons: First, it is guaranteed to yield an expected $O(\log k)$ approximation to the optimal clustering with $k$ centers [4]. Second, it is as simple as Lloyd's algorithm, both conceptually as well as computationally with $O(nkd)$ running time in $d$ dimensions.

The particular focus of this paper is on the setting where an optimal $k$-clustering remains the benchmark but more than $k$ cluster centers can be sampled to improve the approximation. Specifically, it is shown that for any constant factor $\beta > 1$, if $\beta k$ centers are chosen by $D^\ell$ sampling, then a constant-factor approximation to the optimal $k$-clustering is obtained. This guarantee holds in expectation and for all datasets, like the one in [4], and improves upon the $O(\log k)$ factor therein. Such a result is known as a constant-factor bi-criteria approximation since both the optimal cost and the relevant degrees of freedom ($k$ in this case) are exceeded but only by constant factors.

In the context of clustering, bi-criteria approximation guarantees can be valuable because an appropriate number of clusters $k$ is almost never known or pre-specified in practice. Approaches to determining $k$ from the data are ideally based on knowing how the optimal cost decreases as $k$ increases, but obtaining this optimal trade-off between cost and $k$ is NP-hard as mentioned earlier. Alternatively, a simpler algorithm (like $k$-means++) that has a constant-factor bi-criteria guarantee would ensure that the trade-off curve generated by this algorithm deviates by no more than constant factors along both axes from the optimal curve. This may be more appealing than a deviation along

the cost axis that grows as $O(\log k)$. Furthermore, if a solution with a specified number of clusters $k$ is truly required, then linear programming techniques can be used to select a $k$-subset from the $\beta k$ cluster centers while still maintaining a constant-factor approximation [1, 8].

The next section reviews existing work on $D^\ell$ sampling and other clustering approximations. Section 2 formally states the problem, the $D^\ell$ sampling algorithm, and existing lemmas regarding the algorithm. Section 3 states the main results and compares them to previous results. Proofs are presented in Section 4 with more algebraic proofs deferred to the supplementary material.

## 1.1 Related Work

Approximation algorithms for $k$-means ($\ell = 2$), $k$-medians ($\ell = 1$), and related problems span a wide range in the trade-off between tighter approximation factors and lower algorithm complexity. At one end, while exact algorithms [14] and polynomial-time approximation schemes (PTAS) (see [22, 18, 9, 12, 13, 10] and references therein) may have polynomial running times in $n$, the dependence on $k$ and/or the dimension $d$ is exponential or worse. Simpler local search [17, 5] and linear programming [8, 16] algorithms offer constant-factor approximations but still with high-order polynomial running times in $n$, and some rely on dense discretizations of size $O(n\epsilon^{-d}\log(1/\epsilon))$.

In contrast to the above, this paper focuses on highly practical algorithms in the $D^\ell$ sampling class, including $k$-means++. As mentioned, it was proved in [4] that $D^\ell$ sampling results in an $O(\log k)$ approximation, in expectation and for all datasets. The current work extends this guarantee to the constant-factor bi-criteria regime, also for all datasets. The authors of [4] also provided a matching lower bound, exhibiting a dataset on which $k$-means++ achieves an expected $\Omega(\log k)$ approximation.

Improved $O(1)$ approximation factors have been shown for sampling algorithms like $k$-means++ provided that the dataset satisfies certain conditions. Such results were established in [24] for $k$-means++ and other variants of Lloyd's algorithm under the condition that the dataset is well-suited in a sense to partitioning into $k$ clusters, and for an algorithm called successive sampling [23] with $O(n(k + \log n) + k^2 \log^2 n)$ running time subject to a bound on the dispersion of the points.

In a similar direction to the one pursued in the present work, [1] showed that if the number of cluster centers is increased to a constant factor times $k$, then $k$-means++ can achieve a constant-factor approximation, albeit only with constant probability. An $O(1)$ factor was also obtained independently by [2] using more centers, of order $O(k \log k)$. It is important to note that the constant-probability result of [1] in no way implies the main results herein, which are true in expectation and are therefore stronger guarantees. Furthermore, Section 3.1 shows that a constant-probability corollary of Theorem 1 improves upon [1] by more than a factor of 2.

Recently, [21, 7] have also established constant-factor bi-criteria results for the $k$-means problem. These works differ from the present paper in studying more complex local search and linear programming algorithms applied to large discretizations, of size $n^{O(\log(1/\epsilon)/\epsilon^2)}$ (a high-order polynomial) in [21] and $O(n\epsilon^{-d}\log(1/\epsilon))$ in [7], the latter the same as in [17]. Moreover, [7] employs search neighborhoods that are also of exponential size in $d$ (requiring doubly exponential running time).

# 2 Preliminaries

## 2.1 Problem Definition

We are given $n$ points $x_1, \ldots, x_n$ in a real metric space $\mathcal{X}$ with metric $D(x, y)$. The objective is to choose $t$ cluster centers $c_1, \ldots, c_t$ in $\mathcal{X}$ and assign points to the nearest cluster center to minimize the potential function

$$\phi = \sum_{i=1}^{n} \min_{j=1,\ldots,t} D(x_i, c_j)^\ell. \tag{1}$$

A cluster is thus defined by the points $x_i$ assigned to a center $c_j$, where ties (multiple closest centers) are broken arbitrarily. For a subset of points $\mathcal{S}$, define $\phi(\mathcal{S}) = \sum_{x_i \in \mathcal{S}} \min_{j=1,\ldots,t} D(x_i, c_j)^\ell$ to be the contribution to the potential from $\mathcal{S}$; $\phi(x_i)$ is the contribution from a single point $x_i$.

The exponent $\ell \geq 1$ in (1) is regarded as a problem parameter. Letting $\ell = 2$ and $D$ be Euclidean distance, we have what is usually known as the $k$-means problem, so-called because the optimal

**Algorithm 1** $D^\ell$ Sampling
___
**Input:** Data points $x_1, \ldots, x_n$, number of clusters $t$. Initialize $\phi(x_i) = 1$ for $i = 1, \ldots, n$.
**for** $j = 1$ **to** $t$ **do**
    Select $j$th center $c_j = x_i$ with probability $\phi(x_i)/\phi$.
    Update $\phi(x_i)$ for $i = 1, \ldots, n$.
___

cluster centers are means of the points assigned to them. The choice $\ell = 1$ is also popular and corresponds to the $k$-medians problem.

Throughout this paper, an optimal clustering will always refer to one that minimizes (1) over solutions with $t = k$ clusters, where $k \geq 2$ is given. Likewise, the term optimal cluster and symbol $\mathcal{A}$ will refer to one of the $k$ clusters from this optimal solution. The goal is to approximate the potential $\phi^*$ of this optimal $k$-clustering using $t = \beta k$ cluster centers for $\beta \geq 1$.

## 2.2 $D^\ell$ Sampling Algorithm

The $D^\ell$ sampling algorithm chooses cluster centers randomly from $x_1, \ldots, x_n$ with probabilities proportional to their current contributions to the potential, as detailed in Algorithm 1. Following [4], the case $\ell = 2$ is referred to as the $k$-means++ algorithm and the non-uniform probabilities used after the first iteration are referred to as $D^2$ weighting (hence $D^\ell$ in general). For $t$ cluster centers, the running time of $D^\ell$ sampling is $O(ntd)$ in $d$ dimensions.

In practice, Algorithm 1 is used as an initialization to Lloyd's algorithm, which usually produces further decreases in the potential. The analysis herein pertains only to Algorithm 1 and not to the subsequent improvement due to Lloyd's algorithm.

## 2.3 Existing Lemmas Regarding $D^\ell$ Sampling

The following lemmas synthesize useful results from [4] that bound the expected potential within a single optimal cluster due to selecting a center from that cluster with uniform or $D^\ell$ weighting.

**Lemma 1.** *[4, Lemmas 3.1 and 5.1] Given an optimal cluster $\mathcal{A}$, let $\phi$ be the potential resulting from selecting a first cluster center randomly from $\mathcal{A}$ with uniform weighting. Then $\mathbb{E}[\phi(\mathcal{A})] \leq r_u^{(\ell)} \phi^*(\mathcal{A})$ for any $\mathcal{A}$, where*

$$r_u^{(\ell)} = \begin{cases} 2, & \ell = 2 \text{ and } D \text{ is Euclidean}, \\ 2^\ell, & \text{otherwise}. \end{cases}$$

**Lemma 2.** *[4, Lemma 3.2] Given an optimal cluster $\mathcal{A}$ and an initial potential $\phi$, let $\phi'$ be the potential resulting from adding a cluster center selected randomly from $\mathcal{A}$ with $D^\ell$ weighting. Then $\mathbb{E}[\phi'(\mathcal{A})] \leq r_D^{(\ell)} \phi^*(\mathcal{A})$ for any $\mathcal{A}$, where $r_D^{(\ell)} = 2^\ell r_u^{(\ell)}$.*

The factor of $2^\ell$ between $r_u^{(\ell)}$ and $r_D^{(\ell)}$ for general $\ell$ is explained just before Theorem 5.1 in [4].

## 3 Main Results

The main results of this paper are stated below in terms of the single-cluster approximation ratio $r_D^{(\ell)}$ defined by Lemma 2. Subsequently in Section 3.1, the results are discussed in the context of previous work.

**Theorem 1.** *Let $\phi$ be the potential resulting from selecting $\beta k$ cluster centers according to Algorithm 1, where $\beta \geq 1$. The expected approximation ratio is then bounded as*

$$\frac{\mathbb{E}[\phi]}{\phi^*} \leq r_D^{(\ell)} \left( 1 + \min \left\{ \frac{\varphi(k-2)}{(\beta-1)k + \varphi}, H_{k-1} \right\} \right) - \Theta \left( \frac{1}{n} \right),$$

*where $\varphi = (1 + \sqrt{5})/2 \doteq 1.618$ is the golden ratio and $H_k = 1 + \frac{1}{2} + \cdots + \frac{1}{k} \sim \log k$ is the $k$th harmonic number.*

In the proof of Theorem 1 in Section 4.2, it is shown that the $1/n$ term is indeed non-positive and can therefore be omitted, with negligible loss for large $n$.

The approximation ratio bound in Theorem 1 is stated as a function of $k$. The following corollary confirms that the theorem also implies a constant-factor bi-criteria approximation.

**Corollary 1.** *With the same definitions as in Theorem 1, the expected approximation ratio is bounded as*

$$\frac{\mathbb{E}[\phi]}{\phi^*} \leq r_D^{(\ell)} \left(1 + \frac{\varphi}{\beta - 1}\right).$$

*Proof.* The minimum in Theorem 1 is bounded by its first term. This term is in turn increasing in $k$ with asymptote $\varphi/(\beta - 1)$, which can therefore be taken as a $k$-independent bound. □

It follows from Corollary 1 that a constant "oversampling" ratio $\beta > 1$ leads to a constant-factor approximation. Theorem 1 offers a further refinement for finite $k$.

The bounds in Theorem 1 and Corollary 1 consist of two factors. As $\beta$ increases, the second, parenthesized factor decreases to 1 either exactly or approximately as $1/(\beta - 1)$. The first factor of $r_D^{(\ell)}$ however is no smaller than 4, and is a direct consequence of Lemma 2. Any future work on improving Lemma 2 would therefore strengthen the approximation factors above.

## 3.1 Comparisons to Existing Results

A comparison of Theorem 1 to results in [4] is implicit in its statement since the $H_{k-1}$ term in the minimum comes directly from [4, Theorems 3.1 and 5.1]. For $k = 2, 3$, the first term in the minimum is smaller than $H_{k-1}$ for any $\beta \geq 1$, and hence Theorem 1 is always an improvement. For $k > 3$, Theorem 1 improves upon [4] for $\beta$ greater than the critical value

$$\beta_c = 1 + \frac{\phi(k - 2 - H_{k-1})}{k H_{k-1}}.$$

Numerical evaluation of $\beta_c$ shows that it reaches a maximum value of $1.204$ at $k = 22$ and then decreases back toward 1 roughly as $1/H_{k-1}$. It can be concluded that for any $k$, at most 20% oversampling is required for Theorem 1 to guarantee a better approximation than [4].

The most closely related result to Theorem 1 and Corollary 1 is found in [1, Theorem 1]. The latter establishes a constant-factor bi-criteria approximation that holds only with constant probability, as opposed to in expectation. Since a bound on the expectation implies a bound with constant probability via Markov's inequality (but not the other way around), a direct comparison with [1] is possible. Specifically, for $\ell = 2$ and the $t = \lceil 16(k + \sqrt{k}) \rceil$ cluster centers assumed in [1], Theorem 1 in the present work implies that

$$\frac{\mathbb{E}[\phi]}{\phi^*} \leq 8 \left(1 + \min\left\{\frac{\varphi(k - 2)}{\lceil 15k + 16\sqrt{k} \rceil + \varphi}, H_{k-1}\right\}\right) \leq 8 \left(1 + \frac{\varphi}{15}\right),$$

after taking $k \to \infty$. Then by Markov's inequality,

$$\frac{\phi}{\phi^*} \leq \frac{8}{0.97} \left(1 + \frac{\varphi}{15}\right) \doteq 9.137$$

with probability at least $1 - 0.97 = 0.03$ as in [1]. This $9.137$ approximation factor is less than half the factor of 20 in [1].

Corollary 1 may also be compared to the results in [21], which are obtained through more complex algorithms applied to a large discretization, of size $n^{O(\log(1/\epsilon)/\epsilon^2)}$ for reasonably small $\epsilon$. The main difference between Corollary 1 and the bounds in [21] is the extra factor of $r_D^{(\ell)}$. As discussed above, this factor is due to Lemma 2 and is unlikely to be intrinsic to the $D^\ell$ sampling algorithm.

## 4  Proofs

The overall strategy used to prove Theorem 1 is similar to that in [4]. The key intermediate result is Lemma 3 below, which relates the potential at a later iteration in Algorithm 1 to the potential at an earlier iteration. Section 4.1 is devoted to proving Lemma 3. Subsequently in Section 4.2, Theorem 1 is proven by an application of Lemma 3.

In the sequel, we say that an optimal cluster $\mathcal{A}$ is covered by a set of cluster centers if at least one of the centers lies in $\mathcal{A}$. Otherwise $\mathcal{A}$ is uncovered. Also define $\rho = r_D^{(\ell)} \phi^*$ as an abbreviation.

**Lemma 3.** *For an initial set of centers leaving $u$ optimal clusters uncovered, let $\phi$ denote the potential, $\mathcal{U}$ the union of uncovered clusters, and $\mathcal{V}$ the union of covered clusters. Let $\phi'$ denote the potential resulting from adding $t \geq u$ centers, each selected randomly with $D^\ell$ weighting as in Algorithm 1. Then the new potential is bounded in expectation as*

$$\mathbb{E}[\phi' \mid \phi] \leq c_\mathcal{V}(t, u)\phi(\mathcal{V}) + c_\mathcal{U}(t, u)\rho(\mathcal{U})$$

*for coefficients $c_\mathcal{V}(t, u)$ and $c_\mathcal{U}(t, u)$ that depend only on $t, u$. This holds in particular for*

$$c_\mathcal{V}(t, u) = \frac{t + au + b}{t - u + b} = 1 + \frac{(a+1)u}{t - u + b}, \tag{2a}$$

$$c_\mathcal{U}(t, u) = \begin{cases} c_\mathcal{V}(t-1, u-1), & u > 0, \\ 0, & u = 0, \end{cases} \tag{2b}$$

*where the parameters $a$ and $b$ satisfy $a + 1 \geq b > 0$ and $ab \geq 1$. The choice of $a, b$ that minimizes $c_\mathcal{V}(t, u)$ in (2a) is $a + 1 = b = \varphi$.*

## 4.1 Proof of Lemma 3

Lemma 3 is proven using induction, showing that if it holds for $(t, u)$ and $(t, u+1)$, then it also holds for $(t+1, u+1)$, similar to the proof of [4, Lemma 3.3]. The proof is organized into three parts. Section 4.1.1 provides base cases. In Section 4.1.2, sufficient conditions on the coefficients $c_\mathcal{V}(t, u)$, $c_\mathcal{U}(t, u)$ are derived that allow the inductive step to be completed. In Section 4.1.3, it is shown that the closed-form expressions in (2) are consistent with the base cases in Section 4.1.1 and satisfy the sufficient conditions from Section 4.1.2, thus completing the proof.

### 4.1.1 Base cases

This subsection exhibits two base cases of Lemma 3. The first case corresponds to $u = 0$, for which we have $\phi(\mathcal{V}) = \phi$. Since adding centers cannot increase the potential, i.e. $\phi' \leq \phi$ deterministically, Lemma 3 holds with

$$c_\mathcal{V}(t, 0) = 1, \quad c_\mathcal{U}(t, 0) = 0, \quad t \geq 0. \tag{3}$$

The second base case occurs for $t = u$, $u \geq 1$. For this purpose, a slightly strengthened version of [4, Lemma 3.3] is used, as given next.

**Lemma 4.** *With the same definitions as in Lemma 3 except with $t \leq u$, we have*

$$\mathbb{E}[\phi' \mid \phi] \leq (1 + H_t)\phi(\mathcal{V}) + (1 + H_{t-1})\rho(\mathcal{U}) + \frac{u - t}{u}\phi(\mathcal{U}),$$

*where we define $H_0 = 0$ and $H_{-1} = -1$ for convenience.*

The improvement is in the coefficient in front of $\rho(\mathcal{U})$, from $(1 + H_t)$ to $(1 + H_{t-1})$. The proof follows that of [4, Lemma 3.3] with some differences and is deferred to the supplementary material.

Specializing to the case $t = u$, Lemma 4 coincides with Lemma 3 with coefficients

$$c_\mathcal{V}(u, u) = 1 + H_u, \quad c_\mathcal{U}(u, u) = 1 + H_{u-1}. \tag{4}$$

### 4.1.2 Sufficient conditions on coefficients

We now assume inductively that Lemma 3 holds for $(t, u)$ and $(t, u+1)$. The induction to the case $(t+1, u+1)$ is then completed under the following sufficient conditions on the coefficients:

$$c_\mathcal{V}(t, u+1) \geq 1, \tag{5a}$$

$$(c_\mathcal{V}(t, u+1) - c_\mathcal{U}(t, u+1))c_\mathcal{V}(t, u)^2 \geq (c_\mathcal{U}(t, u+1) - c_\mathcal{V}(t, u))^2, \tag{5b}$$

and

$$c_\mathcal{V}(t+1, u+1) \geq \frac{1}{2}\left[c_\mathcal{V}(t, u) + \left(c_\mathcal{V}(t, u)^2 + 4\max\{c_\mathcal{V}(t, u+1) - c_\mathcal{V}(t, u), 0\}\right)^{1/2}\right], \tag{6a}$$

$$c_\mathcal{U}(t+1, u+1) \geq c_\mathcal{V}(t, u). \tag{6b}$$

The first pair of conditions (5) applies to the coefficients involved in the inductive hypothesis for $(t, u)$ and $(t, u + 1)$. The second pair (6) can be seen as a recursive specification of the new coefficients for $(t + 1, u + 1)$. This inductive step together with base cases (3) and (4) are sufficient to extend Lemma 3 to all $t > u$, starting with $(t, u) = (1, 0)$ and $(t, u + 1) = (1, 1)$.

The inductive step is broken down into a series of three lemmas, each building upon the last. The first lemma applies the inductive hypothesis to derive a bound on the potential that depends not only on $\phi(\mathcal{V})$ and $\rho(\mathcal{U})$ but also on $\phi(\mathcal{U})$.

**Lemma 5.** *Assume that Lemma 3 holds for $(t, u)$ and $(t, u + 1)$. Then for the case $(t + 1, u + 1)$, i.e. $\phi$ corresponding to $u + 1$ uncovered clusters and $\phi'$ resulting after adding $t + 1$ centers,*

$$\mathbb{E}[\phi' \mid \phi] \le \min \left\{ \frac{c_\mathcal{V}(t, u)\phi(\mathcal{U}) + c_\mathcal{V}(t, u + 1)\phi(\mathcal{V})}{\phi(\mathcal{U}) + \phi(\mathcal{V})}\phi(\mathcal{V}) \right.$$
$$\left. + \frac{c_\mathcal{V}(t, u)\phi(\mathcal{U}) + c_\mathcal{U}(t, u + 1)\phi(\mathcal{V})}{\phi(\mathcal{U}) + \phi(\mathcal{V})}\rho(\mathcal{U}), \phi(\mathcal{U}) + \phi(\mathcal{V}) \right\}.$$

*Proof.* We consider the two cases in which the first of the $t + 1$ new centers is chosen from either the covered set $\mathcal{V}$ or the uncovered set $\mathcal{U}$. Denote by $\phi^1$ the potential after adding the first new center.

*Covered case:* This case occurs with probability $\phi(\mathcal{V})/\phi$ and leaves the covered and uncovered sets unchanged. We then invoke Lemma 3 with $(t, u + 1)$ (one fewer center to add) and $\phi^1$ playing the role of $\phi$. The contribution to $\mathbb{E}[\phi' \mid \phi]$ from this case is then bounded by

$$\frac{\phi(\mathcal{V})}{\phi} \left( c_\mathcal{V}(t, u + 1)\phi^1(\mathcal{V}) + c_\mathcal{U}(t, u + 1)\rho(\mathcal{U}) \right) \le \frac{\phi(\mathcal{V})}{\phi} \left( c_\mathcal{V}(t, u + 1)\phi(\mathcal{V}) + c_\mathcal{U}(t, u + 1)\rho(\mathcal{U}) \right),$$
(7)

noting that $\phi^1(\mathcal{S}) \le \phi(\mathcal{S})$ for any set $\mathcal{S}$.

*Uncovered case:* We consider each uncovered cluster $\mathcal{A} \subseteq \mathcal{U}$ separately. With probability $\phi(\mathcal{A})/\phi$, the first new center is selected from $\mathcal{A}$, moving $\mathcal{A}$ from the uncovered to the covered set and reducing the number of uncovered clusters by one. Applying Lemma 3 for $(t, u)$, the contribution to $\mathbb{E}[\phi' \mid \phi]$ is bounded by

$$\frac{\phi(\mathcal{A})}{\phi} \left[ c_\mathcal{V}(t, u) \left( \phi^1(\mathcal{V}) + \phi^1(\mathcal{A}) \right) + c_\mathcal{U}(t, u)(\rho(\mathcal{U}) - \rho(\mathcal{A})) \right].$$

Taking the expectation with respect to possible centers in $\mathcal{A}$ and using Lemma 2 and $\phi^1(\mathcal{V}) \le \phi(\mathcal{V})$, we obtain the further bound

$$\frac{\phi(\mathcal{A})}{\phi} \left[ c_\mathcal{V}(t, u)(\phi(\mathcal{V}) + \rho(\mathcal{A})) + c_\mathcal{U}(t, u)(\rho(\mathcal{U}) - \rho(\mathcal{A})) \right].$$

Summing over $\mathcal{A} \subseteq \mathcal{U}$ yields

$$\frac{\phi(\mathcal{U})}{\phi}(c_\mathcal{V}(t, u)\phi(\mathcal{V}) + c_\mathcal{U}(t, u)\rho(\mathcal{U})) + \frac{c_\mathcal{V}(t, u) - c_\mathcal{U}(t, u)}{\phi} \sum_{\mathcal{A} \subseteq \mathcal{U}} \phi(\mathcal{A})\rho(\mathcal{A})$$

$$\le \frac{\phi(\mathcal{U})}{\phi}c_\mathcal{V}(t, u)(\phi(\mathcal{V}) + \rho(\mathcal{U})),$$
(8)

using the inner product bound $\sum_{\mathcal{A} \subseteq \mathcal{U}} \phi(\mathcal{A})\rho(\mathcal{A}) \le \phi(\mathcal{U})\rho(\mathcal{U})$.

The result follows from summing (7) and (8) and combining with the trivial bound $\mathbb{E}[\phi' \mid \phi] \le \phi = \phi(\mathcal{U}) + \phi(\mathcal{V})$. $\square$

The bound in Lemma 5 depends on $\phi(\mathcal{U})$, the potential over uncovered clusters, which can be arbitrarily large or small. In the next lemma, $\phi(\mathcal{U})$ is eliminated by maximizing with respect to it.

**Lemma 6.** *Assume that Lemma 3 holds for $(t, u)$ and $(t, u + 1)$ with $c_\mathcal{V}(t, u + 1) \ge 1$. Then for the case $(t + 1, u + 1)$ in the sense of Lemma 5,*

$$\mathbb{E}[\phi' \mid \phi] \le \frac{1}{2} c_\mathcal{V}(t, u)(\phi(\mathcal{V}) + \rho(\mathcal{U})) + \frac{1}{2} \max \left\{ c_\mathcal{V}(t, u)(\phi(\mathcal{V}) + \rho(\mathcal{U})), \sqrt{Q} \right\},$$

*where*

$$Q = \left(c_\mathcal{V}(t,u)^2 - 4c_\mathcal{V}(t,u) + 4c_\mathcal{V}(t,u+1)\right)\phi(\mathcal{V})^2$$
$$+ 2\left(c_\mathcal{V}(t,u)^2 - 2c_\mathcal{V}(t,u) + 2c_\mathcal{U}(t,u+1)\right)\phi(\mathcal{V})\rho(\mathcal{U}) + c_\mathcal{V}(t,u)^2\rho(\mathcal{U})^2.$$

*Proof.* Let $B_1(\phi(\mathcal{U}))$ and $B_2(\phi(\mathcal{U}))$ denote the two terms inside the minimum in Lemma 5 (i.e. $B_2(\phi(\mathcal{U})) = \phi(\mathcal{U}) + \phi(\mathcal{V})$). The derivative of $B_1(\phi(\mathcal{U}))$ with respect to $\phi(\mathcal{U})$ is given by

$$B_1'(\phi(\mathcal{U})) = \frac{\phi(\mathcal{V})}{(\phi(\mathcal{U}) + \phi(\mathcal{V}))^2}\left[(c_\mathcal{V}(t,u) - c_\mathcal{V}(t,u+1))\phi(\mathcal{V}) + (c_\mathcal{V}(t,u) - c_\mathcal{U}(t,u+1))\rho(\mathcal{U})\right],$$

which does not change sign as a function of $\phi(\mathcal{U})$. The two cases $B_1'(\phi(\mathcal{U})) \geq 0$ and $B_1'(\phi(\mathcal{U})) < 0$ are considered separately below. Taking the maximum of the resulting bounds (9), (10) establishes the lemma.

*Case $B_1'(\phi(\mathcal{U})) \geq 0$:* Both $B_1(\phi(\mathcal{U}))$ and $B_2(\phi(\mathcal{U}))$ are non-decreasing functions of $\phi(\mathcal{U})$. The former has the finite supremum

$$c_\mathcal{V}(t,u)(\phi(\mathcal{V}) + \rho(\mathcal{U})), \tag{9}$$

whereas the latter increases without bound. Therefore $B_1(\phi(\mathcal{U}))$ eventually becomes the smaller of the two and (9) can be taken as an upper bound on $\min\{B_1(\phi(\mathcal{U})), B_2(\phi(\mathcal{U}))\}$.

*Case $B_1'(\phi(\mathcal{U})) < 0$:* At $\phi(\mathcal{U}) = 0$, we have $B_1(0) = c_\mathcal{V}(t,u+1)\phi(\mathcal{V}) + c_\mathcal{U}(t,u+1)\rho(\mathcal{U})$ and $B_2(0) = \phi(\mathcal{V})$. The assumption $c_\mathcal{V}(t,u+1) \geq 1$ implies that $B_1(0) \geq B_2(0)$. Since $B_1(\phi(\mathcal{U}))$ is now a decreasing function, the two functions must intersect and the point of intersection then provides an upper bound on $\min\{B_1(\phi(\mathcal{U})), B_2(\phi(\mathcal{U}))\}$. The supplementary material provides some algebraic details on solving for the intersection. The resulting bound is

$$\frac{1}{2}c_\mathcal{V}(t,u)(\phi(\mathcal{V}) + \rho(\mathcal{U})) + \frac{1}{2}\sqrt{Q}. \tag{10}$$

$\square$

The bound in Lemma 6 is a nonlinear function of $\phi(\mathcal{V})$ and $\rho(\mathcal{U})$, in contrast to the desired form in Lemma 3. The next step is to linearize the bound by imposing additional conditions (5).

**Lemma 7.** *Assume that Lemma 3 holds for $(t,u)$ and $(t,u+1)$ with coefficients satisfying (5). Then for the case $(t+1, u+1)$ in the sense of Lemma 5,*

$$\mathbb{E}[\phi' \mid \phi] \leq \frac{1}{2}\left[c_\mathcal{V}(t,u) + \left(c_\mathcal{V}(t,u)^2 + 4\max\{c_\mathcal{V}(t,u+1) - c_\mathcal{V}(t,u), 0\}\right)^{1/2}\right]\phi(\mathcal{V}) + c_\mathcal{V}(t,u)\rho(\mathcal{U}).$$

*Proof.* It suffices to linearize the $\sqrt{Q}$ term in Lemma 6, specifically by showing that $Q \leq (a\phi(\mathcal{V}) + b\rho(\mathcal{U}))^2$ for all $\phi(\mathcal{V}), \rho(\mathcal{U})$ with $a = \left[c_\mathcal{V}(t,u)^2 + 4(c_\mathcal{V}(t,u+1) - c_\mathcal{V}(t,u))\right]^{1/2}$ and $b = c_\mathcal{V}(t,u)$. Proof of this inequality is provided in the supplementary material. Incorporating the inequality into Lemma 6 proves the result. $\square$

Given conditions (5) and Lemma 7, the inductive step for Lemma 3 can be completed by defining $c_\mathcal{V}(t+1, u+1)$ and $c_\mathcal{U}(t+1, u+1)$ recursively as in (6). $\square$

### 4.1.3  Proof with specific form for coefficients

We now prove that Lemma 3 holds for coefficients $c_\mathcal{V}(t,u)$, $c_\mathcal{U}(t,u)$ given by (2) with $a + 1 \geq b > 0$ and $ab \geq 1$. Given the inductive approach and the results established in Sections 4.1.1 and 4.1.2, the proof requires the remaining steps below. First, it is shown that the base cases (3), (4) from Section 4.1.1 imply that Lemma 3 is true for the same base cases but with $c_\mathcal{V}(t,u)$, $c_\mathcal{U}(t,u)$ given by (2) instead. Second, (2) is shown to satisfy conditions (5) for all $t > u$, thus permitting Lemma 7 to be used. Third, (2) is also shown to satisfy (6), which combined with Lemma 7 completes the induction.

Considering the base cases, for $u = 0$, (3) and (2) coincide so there is nothing to prove. For the case $t = u$, $u \geq 1$, Lemma 3 with coefficients given by (4) implies the same with coefficients given by (2) provided that

$$(1 + H_u)\phi(\mathcal{V}) + (1 + H_{u-1})\rho(\mathcal{U}) \leq \left(1 + \frac{(a+1)u}{b}\right)\phi(\mathcal{V}) + \left(1 + \frac{(a+1)(u-1)}{b}\right)\rho(\mathcal{U})$$

for all $\phi(\mathcal{V})$, $\rho(\mathcal{U})$. This in turn is ensured if the coefficients satisfy $H_u \leq (a+1)u/b$ for all $u \geq 1$. The most stringent case is $u = 1$ and corresponds to the assumption $a + 1 \geq b$.

For the second step of establishing (5), it is clear that (5a) is satisfied by (2a). A direct calculation presented in the supplementary material shows that (5b) is also true.

**Lemma 8.** *Condition* (5b) *is satisfied for all* $t > u$ *if* $c_\mathcal{V}(t, u)$, $c_\mathcal{U}(t, u)$ *are given by* (2) *and* $ab \geq 1$.

Similarly for the third step, it suffices to show that (2a) satisfies recursion (6a) since (2b) automatically satisfies (6b). A proof is provided in the supplementary material.

**Lemma 9.** *Recursion* (6a) *is satisfied for all* $t > u$ *if* $c_\mathcal{V}(t, u)$ *is given by* (2a) *and* $ab \geq 1$.

Lastly, we minimize $c_\mathcal{V}(t, u)$ in (2a) with respect to $a$, $b$, subject to $a + 1 \geq b > 0$ and $ab \geq 1$. For fixed $a$, minimizing with respect to $b$ yields $b = a + 1$ and $c_\mathcal{V}(t, u) = 1 + ((a+1)u)/(t - u + a + 1)$. Minimizing with respect to $a$ then results in setting $ab = a(a+1) = 1$. The solution satisfying $a + 1 > 0$ is $a = \varphi - 1$ and $b = \varphi$.

## 4.2  Proof of Theorem 1

Denote by $n_\mathcal{A}$ the number of points in optimal cluster $\mathcal{A}$. In the first iteration of Algorithm 1, the first cluster center is selected from some $\mathcal{A}$ with probability $n_\mathcal{A}/n$. Conditioned on this event, Lemma 3 is applied with covered set $\mathcal{V} = \mathcal{A}$, $u = k - 1$ uncovered clusters, and $t = \beta k - 1$ remaining cluster centers. This bounds the final potential $\phi'$ as

$$\mathbb{E}[\phi' \mid \phi] \leq c_\mathcal{V}(\beta k - 1, k - 1)\phi(\mathcal{A}) + c_\mathcal{U}(\beta k - 1, k - 1)(\rho - \rho(\mathcal{A}))$$

where $c_\mathcal{V}(t, u)$, $c_\mathcal{U}(t, u)$ are given by (2) with $a + 1 = b = \varphi$. Taking the expectation over possible centers in $\mathcal{A}$ and using Lemma 1,

$$\mathbb{E}[\phi' \mid \mathcal{A}] \leq r_u^{(\ell)} c_\mathcal{V}(\beta k - 1, k - 1)\phi^*(\mathcal{A}) + c_\mathcal{U}(\beta k - 1, k - 1)(\rho - \rho(\mathcal{A})).$$

Taking the expectation over clusters $\mathcal{A}$ and recalling that $\rho = r_D^{(\ell)}\phi^*$,

$$\mathbb{E}[\phi'] \leq r_D^{(\ell)} c_\mathcal{U}(\beta k - 1, k - 1)\phi^* - C \sum_\mathcal{A} \frac{n_\mathcal{A}}{n}\phi^*(\mathcal{A}), \tag{11}$$

where $C = r_D^{(\ell)} c_\mathcal{U}(\beta k - 1, k - 1) - r_u^{(\ell)} c_\mathcal{V}(\beta k - 1, k - 1)$. Using (2) and $r_D^{(\ell)} = 2^\ell r_u^{(\ell)}$ from Lemma 2,

$$C = r_u^{(\ell)} \frac{2^\ell ((\beta - 1)k + \varphi(k - 1)) - (\beta - 1 + \varphi)k}{(\beta - 1)k + \varphi}$$

$$= r_u^{(\ell)} \frac{(2^\ell - 1)(\beta - 1)k + \varphi((2^\ell - 1)(k - 1) - 1)}{(\beta - 1)k + \varphi}.$$

The last expression for $C$ is seen to be non-negative for $\beta \geq 1$, $k \geq 2$, and $\ell \geq 1$. Furthermore, since $n_\mathcal{A} = 1$ (a singleton cluster) implies that $\phi^*(\mathcal{A}) = 0$, we have

$$\sum_\mathcal{A} n_\mathcal{A}\phi^*(\mathcal{A}) = \sum_{\mathcal{A}: n_\mathcal{A} \geq 2} n_\mathcal{A}\phi^*(\mathcal{A}) \geq 2\phi^*. \tag{12}$$

Substituting (2) and (12) into (11), we obtain

$$\frac{\mathbb{E}[\phi']}{\phi^*} \leq r_D^{(\ell)}\left(1 + \frac{\varphi(k - 2)}{(\beta - 1)k + \varphi}\right) - \frac{2C}{n}. \tag{13}$$

The last step is to recall [4, Theorems 3.1 and 5.1], which together state that

$$\frac{\mathbb{E}[\phi']}{\phi^*} \leq r_D^{(\ell)}(1 + H_{k-1}) \tag{14}$$

for $\phi'$ resulting from selecting exactly $k$ cluster centers. In fact, (14) also holds for $\beta k$ centers, $\beta \geq 1$, since adding centers cannot increase the potential. The proof is completed by taking the minimum of (13) and (14).

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
