[Supplementary Material]


## Supplementary Material for "A Constant Factor Bi-Criteria Approximation Guarantee for $k$-means++"

The following appendices provide proofs omitted from the main manuscript. Equation numbers and other labels refer back to the main manuscript.

## A  Proof of Lemma 4

The proof follows the inductive proof of [4, Lemma 3.3] with the notational changes $\mathcal{X}_u \to \mathcal{U}$, $\mathcal{X}_c \to \mathcal{V}$, and $8\phi_{\mathrm{OPT}} \to \rho$. For brevity, only the differences are presented.

For the first base case $t = 0$, $u > 0$, [4] already shows that the lemma holds with coefficients $1 = 1 + H_0$, $0 = 1 + H_{-1}$, and $1 = (u - 0)/u$. Similarly for the second base case $t = u = 1$, [4] shows that $\mathbb{E}[\phi' \mid \phi] \leq 2\phi(\mathcal{V}) + \rho(\mathcal{U}) = (1 + H_1)\phi(\mathcal{V}) + (1 + H_0)\rho(\mathcal{U})$, as required for the stronger version here.

For the first "covered" case considered in the inductive step, the argument is the same and the upper bound on the contribution to $\mathbb{E}[\phi' \mid \phi]$ is changed to

$$\frac{\phi(\mathcal{V})}{\phi}\left[(1 + H_{t-1})\phi(\mathcal{V}) + (1 + H_{t-2})\rho(\mathcal{U}) + \frac{u - t + 1}{u}\phi(\mathcal{U})\right]. \tag{15}$$

For the second "uncovered" case, the first displayed expression in the right-hand column of [4, page 1030] becomes (after applying the bound $\sum_{a \in \mathcal{A}} p_a \phi_a \leq \rho(\mathcal{A})$ from Lemma 2)

$$\frac{\phi(\mathcal{A})}{\phi}\left[(1 + H_{t-1})(\phi(\mathcal{V}) + \rho(\mathcal{A})) + (1 + H_{t-2})(\rho(\mathcal{U}) - \rho(\mathcal{A})) + \frac{u - t}{u - 1}(\phi(\mathcal{U}) - \phi(\mathcal{A}))\right].$$

Summing over all uncovered clusters $\mathcal{A} \subseteq \mathcal{U}$, the contribution to $\mathbb{E}[\phi' \mid \phi]$ is bounded from above by

$$\frac{\phi(\mathcal{U})}{\phi}\left[(1 + H_{t-1})\phi(\mathcal{V}) + (1 + H_{t-2})\rho(\mathcal{U}) + \frac{u - t}{u - 1}\phi(\mathcal{U})\right]$$
$$+ \frac{1}{\phi}\left[(H_{t-1} - H_{t-2})\sum_{\mathcal{A} \subseteq \mathcal{U}}\phi(\mathcal{A})\rho(\mathcal{A}) - \frac{u - t}{u - 1}\sum_{\mathcal{A} \subseteq \mathcal{U}}\phi(\mathcal{A})^2\right].$$

The inner product above can be bounded as

$$\sum_{\mathcal{A} \subseteq \mathcal{U}}\phi(\mathcal{A})\rho(\mathcal{A}) \leq \phi(\mathcal{U})\rho(\mathcal{U}), \tag{16}$$

with equality if both $\phi(\mathcal{U})$, $\rho(\mathcal{U})$ are completely concentrated in the same cluster $\mathcal{A}$. The sum of squares term can be bounded using the power-mean inequality as in [4]. Hence the contribution to $\mathbb{E}[\phi' \mid \phi]$ is further bounded by

$$\frac{\phi(\mathcal{U})}{\phi}\left[(1 + H_{t-1})\phi(\mathcal{V}) + (1 + H_{t-1})\rho(\mathcal{U}) + \frac{u - t}{u}\phi(\mathcal{U})\right]. \tag{17}$$

Summing the bounds in (15), (17), we have

$$\mathbb{E}[\phi' \mid \phi] \leq (1 + H_{t-1})\phi(\mathcal{V}) + \left(1 + \frac{\phi(\mathcal{V})H_{t-2} + \phi(\mathcal{U})H_{t-1}}{\phi}\right)\rho(\mathcal{U}) + \frac{u - t}{u}\phi(\mathcal{U}) + \frac{\phi(\mathcal{V})}{\phi}\frac{\phi(\mathcal{U})}{u}.$$

Recalling that $\phi = \phi(\mathcal{V}) + \phi(\mathcal{U})$, the right-hand side is seen to be increasing in $\phi(\mathcal{U})$. Taking the worst case as $\phi(\mathcal{U}) \to \phi$ gives

$$\mathbb{E}[\phi' \mid \phi] \leq \left(1 + H_{t-1} + \frac{1}{u}\right)\phi(\mathcal{V}) + (1 + H_{t-1})\rho(\mathcal{U}) + \frac{u - t}{u}\phi(\mathcal{U})$$
$$\leq (1 + H_t)\phi(\mathcal{V}) + (1 + H_{t-1})\rho(\mathcal{U}) + \frac{u - t}{u}\phi(\mathcal{U})$$

since $1/u \leq 1/t$. This completes the induction.

## B  Remainder of Proof of Lemma 6

This appendix provides some additional details on solving for the intersection between the functions

$$B_1(\phi(\mathcal{U})) = \frac{c_\mathcal{V}(t,u)\phi(\mathcal{U}) + c_\mathcal{V}(t,u+1)\phi(\mathcal{V})}{\phi(\mathcal{U}) + \phi(\mathcal{V})}\phi(\mathcal{V}) + \frac{c_\mathcal{V}(t,u)\phi(\mathcal{U}) + c_\mathcal{U}(t,u+1)\phi(\mathcal{V})}{\phi(\mathcal{U}) + \phi(\mathcal{V})}\rho(\mathcal{U}),$$

$$B_2(\phi(\mathcal{U})) = \phi(\mathcal{U}) + \phi(\mathcal{V})$$

for the case that $B_1(\phi(\mathcal{U}))$ is decreasing in $\phi(\mathcal{U})$. Equating $B_1(\phi(\mathcal{U}))$ and $B_2(\phi(\mathcal{U}))$ leads after some algebra to a quadratic equation in $\phi(\mathcal{U})$:

$$0 = \phi(\mathcal{U})^2 + [2\phi(\mathcal{V}) - c_\mathcal{V}(t,u)(\phi(\mathcal{V}) + \rho(\mathcal{U}))]\,\phi(\mathcal{U})$$
$$+ \phi(\mathcal{V})\left(\phi(\mathcal{V}) - c_\mathcal{V}(t,u+1)\phi(\mathcal{V}) - c_\mathcal{U}(t,u+1)\rho(\mathcal{U})\right).$$

By the assumption $c_\mathcal{V}(t,u+1) \geq 1$, the constant term in this quadratic equation is non-positive, implying that one of the roots is also non-positive and can be discarded. The remaining positive root is given by

$$\phi(\mathcal{U}) = \frac{1}{2}c_\mathcal{V}(t,u)(\phi(\mathcal{V}) + \rho(\mathcal{U})) - \phi(\mathcal{V}) + \frac{1}{2}\sqrt{Q}$$

after simplifying the discriminant to match the stated expression for $Q$. Evaluating either $B_1(\phi(\mathcal{U}))$ or $B_2(\phi(\mathcal{U}))$ at this root yields the bound in (10), as required.

## C  Proof of Lemma 7

We aim to bound the quadratic function $Q$ from above by the square $(a\phi(\mathcal{V}) + b\rho(\mathcal{U}))^2$ for all $\phi(\mathcal{V}), \rho(\mathcal{U})$ and some choice of $a, b \geq 0$. The cases $\phi(\mathcal{V}) = 0$ and $\rho(\mathcal{U}) = 0$ require that

$$a^2 \geq c_\mathcal{V}(t,u)^2 + 4(c_\mathcal{V}(t,u+1) - c_\mathcal{V}(t,u)),$$
$$b^2 \geq c_\mathcal{V}(t,u)^2.$$

Setting these inequalities to equalities, the remaining condition for the cross-term is

$$ab \geq c_\mathcal{V}(t,u)^2 + 2(c_\mathcal{U}(t,u+1) - c_\mathcal{V}(t,u)).$$

Equivalently for $a, b \geq 0$,

$$a^2 b^2 = \left(c_\mathcal{V}(t,u)^2 + 4(c_\mathcal{V}(t,u+1) - c_\mathcal{V}(t,u))\right)c_\mathcal{V}(t,u)^2$$
$$\geq \left(c_\mathcal{V}(t,u)^2 + 2(c_\mathcal{U}(t,u+1) - c_\mathcal{V}(t,u))\right)^2.$$

We rearrange to obtain

$$4(c_\mathcal{V}(t,u+1) - c_\mathcal{V}(t,u))c_\mathcal{V}(t,u)^2$$
$$\geq 4c_\mathcal{V}(t,u)^2(c_\mathcal{U}(t,u+1) - c_\mathcal{V}(t,u)) + 4(c_\mathcal{U}(t,u+1) - c_\mathcal{V}(t,u))^2,$$

$$(c_\mathcal{V}(t,u+1) - c_\mathcal{U}(t,u+1))c_\mathcal{V}(t,u)^2 \geq (c_\mathcal{U}(t,u+1) - c_\mathcal{V}(t,u))^2,$$

the last of which is true by assumption (5). Thus we conclude that

$$\sqrt{Q} \leq \sqrt{c_\mathcal{V}(t,u)^2 + 4(c_\mathcal{V}(t,u+1) - c_\mathcal{V}(t,u))}\phi(\mathcal{V}) + c_\mathcal{V}(t,u)\rho(\mathcal{U}).$$

Combining this last inequality with Lemma 6 proves the result.

## D  Proof of Lemma 8

Substituting (2) into the left-most factor in (5b),

$$c_\mathcal{V}(t,u+1) - c_\mathcal{U}(t,u+1) = c_\mathcal{V}(t,u+1) - c_\mathcal{V}(t-1,u)$$
$$= \frac{(a+1)(u+1)}{t-u-1+b} - \frac{(a+1)u}{t-1-u+b}$$
$$= \frac{a+1}{t-u-1+b}.$$

Similarly on the right-hand side of (5b),

$$c_\mathcal{U}(t, u+1) - c_\mathcal{V}(t, u) = c_\mathcal{V}(t-1, u) - c_\mathcal{V}(t, u)$$

$$= \frac{(a+1)u}{t-1-u+b} - \frac{(a+1)u}{t-u+b}$$

$$= \frac{(a+1)u}{(t-u+b)(t-u-1+b)}.$$

Hence

$$(c_\mathcal{V}(t, u+1) - c_\mathcal{U}(t, u+1))c_\mathcal{V}(t, u)^2 - (c_\mathcal{U}(t, u+1) - c_\mathcal{V}(t, u))^2$$

$$= \frac{a+1}{t-u-1+b}\left(1 + 2\frac{(a+1)u}{t-u+b} + \frac{(a+1)^2 u^2}{(t-u+b)^2}\right) - \frac{(a+1)^2 u^2}{(t-u+b)^2(t-u-1+b)^2}$$

$$= \frac{a+1}{t-u-1+b}\left(1 + 2\frac{(a+1)u}{t-u+b}\right) + \frac{(a+1)^2 u^2 \left[(a+1)(t-u-1+b) - 1\right]}{(t-u+b)^2(t-u-1+b)^2}. \quad (18)$$

The first of the two summands in (18) is positive for $t > u \geq 0$. The second summand is also non-negative as long as $(a+1)(t-u-1+b) \geq 1$. The most stringent case occurs for $t = u+1$ and is implied by the assumption $ab \geq 1$. We conclude that (18) is positive, i.e. (5b) holds.

## E   Proof of Lemma 9

First note that (2a) has the property that $c_\mathcal{V}(t, u+1) \geq c_\mathcal{V}(t, u)$ for all $t, u$. Therefore (6a) is equivalent to

$$2c_\mathcal{V}(t+1, u+1) - c_\mathcal{V}(t, u) \geq \sqrt{c_\mathcal{V}(t, u)^2 + 4(c_\mathcal{V}(t, u+1) - c_\mathcal{V}(t, u))}. \quad (19)$$

Substituting (2a) into the left-hand side,

$$2c_\mathcal{V}(t+1, u+1) - c_\mathcal{V}(t, u) = 1 + 2\frac{(a+1)(u+1)}{t-u+b} - \frac{(a+1)u}{t-u+b}$$

$$= 1 + \frac{(a+1)(u+2)}{t-u+b},$$

which is seen to be positive for $t > u \geq 0$. Hence (19) is in turn equivalent to

$$(2c_\mathcal{V}(t+1, u+1) - c_\mathcal{V}(t, u))^2 \geq c_\mathcal{V}(t, u)^2 + 4(c_\mathcal{V}(t, u+1) - c_\mathcal{V}(t, u)).$$

On the left-hand side,

$$(2c_\mathcal{V}(t+1, u+1) - c_\mathcal{V}(t, u))^2 = 1 + 2\frac{(a+1)(u+2)}{t-u+b} + \frac{(a+1)^2(u+2)^2}{(t-u+b)^2}. \quad (20)$$

On the right-hand side,

$$c_\mathcal{V}(t, u+1) - c_\mathcal{V}(t, u) = \frac{(a+1)(u+1)}{t-u-1+b} - \frac{(a+1)u}{t-u+b}$$

$$= \frac{(a+1)(t+b)}{(t-u+b)(t-u-1+b)}$$

$$= \frac{a+1}{t-u+b}\left(1 + \frac{u+1}{t-u-1+b}\right),$$

$$c_\mathcal{V}(t, u)^2 = 1 + 2\frac{(a+1)u}{t-u+b} + \frac{(a+1)^2 u^2}{(t-u+b)^2},$$

$$c_\mathcal{V}(t, u)^2 + 4(c_\mathcal{V}(t, u+1) - c_\mathcal{V}(t, u))$$

$$= 1 + 2\frac{(a+1)(u+2)}{t-u+b} + \frac{(a+1)^2 u^2}{(t-u+b)^2} + 4\frac{(a+1)(u+1)}{(t-u+b)(t-u-1+b)}. \quad (21)$$

386 Subtracting (21) from (20) yields

$$\frac{4(a+1)^2(u+1)}{(t-u+b)^2} - 4\frac{(a+1)(u+1)}{(t-u+b)(t-u-1+b)}$$
$$= 4\frac{(a+1)(u+1)\left[a(t-u-1+b)-1\right]}{(t-u+b)^2(t-u-1+b)},$$

387 which is non-negative provided that $a(t-u-1+b) \geq 1$. As in the proof of Lemma 8, the most
388 stringent case occurs for $t = u + 1$ and is covered by the assumption $ab \geq 1$. We conclude that (20)
389 is at least as large as (21), i.e. (6a) holds.