[Reviews · NeurIPS 2016]

Reviewer 1

Summary

The paper considers D^\ell sampling for the k-means objective. The main result is that if one picks \beta k centers via D^\ell sampling for a constant \beta, the expected cost of the obtained solution is at most a constant times the optimum cost with k centers. (In other words, it is a bicriteria approximation, in expectation.) The paper also gives a tradeoff of \beta vs the expected approximation ratio. Earlier, such a result was known, except with only a constant success probability. The bound obtained here is also better in terms of the dependence of the approximation factor on \beta on the approximation factor.

Qualitative Assessment

The paper provides a careful analysis of D^\ell sampling, when one picks \beta k centers, for k-means clustering. The bounds obtained hold in expectation, and are numerically better than similar bounds that were known earlier. The techniques are also simple, and could apply to the analysis of distance-sampling algorithms for other problems. The algorithm is also a frequently used one, so I believe it is a very good fit for the conference.

Confidence in this Review

2-Confident (read it all; understood it all reasonably well)


Reviewer 2

Summary

The paper shows that a popular clustering heuristic - k-means++ - gives a constant bi-criteria approximation i.e. the algorithm finds beta k centers (instead of k; here beta > 1) such that the cost of k-means is upper bounded by C OPT(k), where OPT(k) is the cost of the optimal solution on k centers. Previously, it was known that k-means++ finds a constant bi-criteria approximation with a constant probability. So one could repeat the algorithm several times to obtain a constant approximation in expectation. However, the result was not known for k-means++ as is. The best non-bi-criteria approximation factor for k-means++ is O(log k).

Qualitative Assessment

k-means++ is a very important heuristic used in practice. While this result is somewhat incremental (cf. [1]), I believe it is important enough to be accepted to NIPS.

Confidence in this Review

2-Confident (read it all; understood it all reasonably well)


Reviewer 3

Summary

This paper extends the analysis of the k-means++ algorithm. It shows a bi-criteria approximation of the following nature: If ck centers (for constant c > 1) are picked using the sampling technique of k-means++, then the solution is a constant (pseudo)approximation solution (i.e., the cost if compared with the optimal cost w.r.t. k centers).

Qualitative Assessment

The results may also be interpreted as a tradeoff between the number of points sampled and the approximation guarantee obtained. I did not go through the details of the proofs (some of which are in the appendix) but they seem to follow the analysis similar to that in the Arthur-Vassilvitskii paper. I think the results in the paper are nice and extends our knowledge about the k-means++ technique. Some specific comments are given below: 1. [Line 5] Claims to extend earlier known O(log k)-approximation for \beta = 1. I think that the results in this paper hold only for \beta > 1. 2. [Line 20] Claims that D^l sampling gives O(log k)-approximation in expectation. It would be better if you mentioned l = 2. 3. [Theorem 1] Mentions \beta > = 1, where as the results seems to be valid for \beta > 1. 4. [Lemma 6] The differentiation part is not clear. What are B_1(\phi(U)) and B_2(\phi(U)), and they are differentiating w.r.t. to which variable? Some clarity will be appreciated.

Confidence in this Review

2-Confident (read it all; understood it all reasonably well)


Reviewer 4

Summary

The authors consider the famous k-means problem; Given a set of points P in a metric space with distance function d, the goal is to output k centers such that the sum of the squared distances of each point to its closest center is minimized. The authors focus on bi-criteria approximation algorithms; The algorithms are allowed to output beta k centers (instead of k) and the approximation ratio is computed with respect to the optimal solution (that uses k centers). In the general case, the optimal solution using k centers can be arbitrarily worse than the solution using beta k centers for beta > 1. The goal of the paper is to prove that a algorithms of a certain kind -- called D-sampling algorithms -- produce constant factor bi-criteria approximation. D-sampling algorithms iteratively pick the set of centers at random according to some probability distribution based on the cost of the points with respect to the current set of centers. This work is directly in the line of D. Arthur and S. Vassilvitskii [4] whose proved that the D-sampling algorithms produce an O(log k)-approximation (not bi-criteria). The main result of this work is to analyze the D-sampling algorithms in the bi-criteria regime. More concretely, they show an approximation guarantee of 1+min{O((k-2)/(beta-1)k), log k} for D-sampling algorithms allowing beta k centers.

Qualitative Assessment

I find the result interesting and it is a nice addition to the literature but I do not find it very surprising in itself. Intuitively, D-sampling algorithms do not need to know k in advance, thus, if the algorithm is efficient for finding k centers (which we know it is from [4]), allowing it to pick beta k centers "has to" make the cost decrease significantly (as if the algorithm was solving an instance of the problem for beta k centers). Moreover, the result was known in the constant probability case (rather than expectation). One of the main issue of this paper is the lack of comparison with previous work. There are at least 4 recent papers that tackle the k-means problem (and two of them tackle the bi-criteria version) and 3 of them are not cited (all on arxiv): 1-- A bi-criteria approximation algorithm for k Means. [20] 2-- On Variants of k-means Clustering. 3-- Local search yields approximation schemes for k-means and k-median in Euclidean and minor-free metrics. 4-- Local Search Yields a PTAS for $k$-Means in Doubling Metrics. How does it compare to the recent bi-criteria approximation algorithms of 1 and 2? Does D-sampling algorithms provide a better bi-criteria approximation? It is stated that the algorithms in 1 have dependency in n^O(log(1/epsilon)/epsilon^2). However, local search with neighborhood of size 1 has running time O(n k) (and constant factor approximation guarantee) which is comparable to D-sampling algorithms. Thus, according to the sentence "The main difference between Corollary 1 and the bounds in [20] is the extra factor of r_D^l." it seens that D-sampling is worse. Another issue is about the motivation of this work. Are bi-criteria D-sampling algorithms used in practice? This is true for not bi-criteria versions, are there references for the bi-criteria case? One interesting question is whether allowing an additional number of clusters, say k + c, would be sufficient to significantly improve the approximation ratio of log(k) (this is true for the case of k-median for example). Other relevant remark: "Alternatively, a simpler algorithm (like k-means++) that has a constant-factor bi-criteria guarantee would ensure that the trade-off curve generated by this algorithm deviates by no more than constant factors along both axes from the optimal curve." It is shown in 2 that local search (which is rather simple) is such an algorithm and this holds even when centers are restricted to be input points (at the cost of a constant factor).

Confidence in this Review

2-Confident (read it all; understood it all reasonably well)


Reviewer 5

Summary

In this paper, a constant factor approximation guaranty is proposed for k-means++ clustering. This approximation is obtained by choosing the initial cluster centers using $D^l$ sampling. Two criteria are considered which a constant factor approximation are guaranteed for both of them: 1) the number of clusters, 2) objective function of k-means which is the sum of the minimum distance of each point to cluster centers.

Qualitative Assessment

Some proofs need more explanation to be easily understandable. Despite the proposed theoretical guaranty which is novel and interesting, lack of experimental evaluation is a weakness for this paper.

Confidence in this Review

2-Confident (read it all; understood it all reasonably well)


Reviewer 6

Summary

The class of D^{\ell} sampling algorithms is one of the most widely used for the classical k-means problem. It was previously known that this class of algorithms provides a O(log(k)) approximation [4] to the optimal clustering with k centers. On the other hand, it was also previously known that some algorithms can achieve a constant bi-criteria approximation [20], i.e. that with a constant violation on the number of centers, one can achieve a constant approximation of the optimal cost. This paper shows that the class of D^{\ell} sampling algorithms also achieves such a bi-criteria approximation.

Qualitative Assessment

This type of result was already known to be achievable ([20] for instance). The contribution of this paper is to show that the well-known k-means++ algorithm also achieves this guarantee. Given that this algorithm is so widely used, this is a good thing to know that the algorithm also offers this guarantee. Moreover, Section 3.1 does a good job at relating the result to existing literature and showing that the result also improve on the constant factor itself. However, it would be nice to see some numerical experiments to illustrate: (1) the natural tradeoff between the two criteria (how does the approximation decreases as beta increases), (2) how the algorithm compare to other existing bi criteria approximation algorithms. From a technical point of view, the paper interestingly generalizes some technical lemmas introduced in [4]. Minor comments: - I personally would state Corollary 1 as Theorem 1 since it is the main contribution of the paper and the comparison to the existing bound of [4] is well discussed in Section 3.1. - I think the proof would be easier to read with the sufficient conditions in Section 4.1.2. being introduced as the proof unfolds and maybe summarized at the end of the section instead of them all being presented upfront.

Confidence in this Review

2-Confident (read it all; understood it all reasonably well)


Reviewer 7

Summary

The paper shows that the kmeans++ algorithm (and in general, the D^\ell family of algorithms) provides a constant factor approximation in expectation when \beta*k cluster centers are chosen.

Qualitative Assessment

The paper comprises of multiple small contributions. However for adaptive sampling techniques for kmeans, these small contributions are interesting. Key contributions: 1. Improves the constant factor in the bi-criteria approximation guarantee. 2. The result holds in expectation rather than with constant probability. 3. The result holds for a general family of algorithms (of which kmeans++ is a member) Remarks: 1. Section 3.1 is useful in showing relation to previous results. 2. The paper is well written and easy to understand.

Confidence in this Review

2-Confident (read it all; understood it all reasonably well)